# The Current Diversity and Distribution of the Simple Thalloid Genus *Apopellia* (Marchantiophyta): Evidence from an Integrative Taxonomic Study

Nadezhda A. Konstantinova [1,*], Anna A. Vilnet [1] and Yuriy S. Mamontov [1,2]

1 Polar-Alpine Botanical Garden-Institute of N.A. Avrorin, Kola Science Centre of Russian Academy of Sciences, 184256 Kirovsk, Russia; anya_v@list.ru (A.A.V.); yur-mamontov@yandex.ru (Y.S.M.)
2 Tsitsin Main Botanical Garden, Russian Academy of Sciences, Botanicheskaja 4, 127276 Moscow, Russia
* Correspondence: nadya50@list.ru

**Abstract:** An integrative study of expanded sampling of *Apopellia* species, including the topotype of *Apopellia megaspora*, made it possible to clarify the taxonomic position and distribution of the species of the genus. The ITS1-2 and *trn*L-F sequence data were obtained for 36 molecularly tested specimens, including the topotype *Apopellia megaspora*, that together with data previously deposited in GenBank, support the treatment of *Apopellia* as a separate genus and *A. alpicola* as a distinct species, as well as radically change the idea on the distribution of the species of the genus. It is shown that *A. megaspora* is an American-Asian species with single records in Europe, whereas *A. alpicola* is a West-American-Eurasian species widespread in western North America and occurring scattered in Eurasia. Both species occur in the mountains of western North America and south Siberia. *A. endiviifolia* is widespread in Europe, scattered in Asia and so far not confirmed for America. The expanded sampling of *Apopellia* spp. allows us to clarify the morphological features of the species of the genus, and microphotographs illustrate the more-important morphological features.

**Keywords:** liverworts; *Apopellia*; molecular analyses; morphology; phylogeny; systematics; distribution; Eurasia; America

## 1. Introduction

The oligotypic genus *Pellia* sensu Müller [1] is widely distributed in the cold and temperate regions of the Holarctic. Three taxa were recognized almost until the middle of the last century: *Pellia endiviifolia* (Dicks.) Dumort. (=*Pellia fabroniana* Raddi); *P. neesiana* (Gottsche) Limpr.; and *P epiphylla* (L.) Corda. However, near the middle and in the second half of the 20th century, several additional species were described due to thorough morphological studies, namely *Pellia borealis* Lorb. from Europe, [2,3], *Pellia columbiana* Krajina et Brayshaw [4] and *Pellia megaspora* R.M. Schust. [5–7] from North America. Later, the heterogeneity within *Pellia* s.l. was revealed in the study of karyotype organization [8] and peroxidase isozyme polymorphism [9]. In combination with morphological distinctions, this allowed the subgenus *Apopellia* Grolle [10] to be segregated within the genus *Pellia*.

The genus *Apopellia* (Grolle) Nebel & D.Quandt was separated from the genus *Pellia* Raddi by Schütz et al. [11] after integrative study with three chloroplast DNA markers of this ancient thalloid taxon. Plants with relatively long polymorphous slime hairs near the apex on the ventral side, thalli in cross section without thickening bands, perichaetia forming a complete ring and ciliate-laciniate mouth were previously considered as section *Pellia* [7] or as subgenus *Apopellia* [10]. In addition to the well-known *A. endiviifolia* (Dicks.) Nebel & D. Quandt, the genus includes *A. megaspora* (R.M. Schust.) Nebel & D. Quandt described by Schuster [5] as *Pellia megaspora* R.M. Schust. from Massachusetts (eastern North America) and *Apopellia alpicola* (R.M. Schust. ex L. Söderstr., A. Hagborg & von Konrat) Nebel & D. Quandt, described by Schuster as "*Pellia endiviifolia* subsp. *alpicola*"

R.M. Schust. (nom. inval.), from western North America [6]. All American specimens used in the molecular study by Schütz et al. [11] were attributed to *A. megaspora* despite the fact that the East American and West American specimens were split on the constructed trees into two subclades with ambiguously resolved relation [11].

Morphological differences, at least for *A. megaspora*, are quite clearly expressed in the spores, hence the name of this species. The problem is that sporophytes in these species, as in all liverworts, are ephemeral, and rarely present in collected specimens stored in herbaria, while the vast majority of specimens are male or sterile, thus, according to the figurative expression of Schuster [5], are "taxonomically meaningless". Therefore, many herbarium specimens of the genus cannot be accurately identified. This problem can now be solved with the help of molecular genetic studies, which requires the barcoding of type species or specimens collected at the type locality.

The main aim of this paper is to clarify the phylogeny and taxonomy of the genus *Apopellia*, as well as distribution of its species. One of the challenges was to sequence specimens of *Apopellia megaspora* collected by the first author during an excursion with R. Schuster to Conway (Massachusetts, Schuster's privite area in Green River Valley), from where Schuster described this species, and thus obtain a DNA loci of *A. megaspora* from the site shown by Schuster as the type locality. Another goal was to consider how to treat the Western North American specimens, which, based on their distribution, ecology and morphology, can be attributed to *A. alpicola*. A further task was to find out whether *A. megaspora* really occurs in the Asian part of Russia, as maintained by Schütz et al. [11], in which the sequenced specimen from Turukhansk, valley of the Yenisei River Krasnoyarsk Territory (not Lena, where the authors mistakenly placed it), turns out to be in one of the subclades of *A. megaspora*, along with Western North American specimens. Finally, we clarify the global distribution of the genus.

## 2. Material and Methods

### 2.1. Collections and Specimens Studied

Specimens of *Pellia endiviifolia* stored in the herbarium of the Polar-Alpine Botanical Garden Institute, Kirovsk, Russia (KPABG) and the herbarium of the Main Botanical Garden, Moscow (MHA) were revised. We studied a total of about 100 specimens from the genus *Apopellia* in KPABG and 98 specimens in MHA, including the topotype of *Apopellia megaspora* preserved in KPABG. In addition, we studied specimens of *Pellia* spp. preserved in KPABG and MHA in order to correct the misidentifications and to reveal the distribution of the *Apopellia* species. Data on selected specimens studied are given in the species descriptions.

### 2.2. Morphological Study

Morphological descriptions of the species are given mainly on the basis of the sequenced specimens completed by published data in square brackets. The specimens were studied and photographed using light microscopes equipped with digital cameras. Plants with perianth and sporophytes were studied in detail. Details of morphological characters of male and female plants, as well as sporophytes, were photographed.

### 2.3. Ecology and Distribution

Information about the ecology and distribution of *Apopellia endiviifolia* is based mostly on studied specimens from KPABG and MHA and critically evaluated data available from labels in L., formerly CRIS (http:// kpabg.ru/h/ accessed on 23 July 2023). Distribution of *A. megaspora* and *A. alpicola* is based on specimens sequenced and revised in this study, with the addition of published data, mostly by Schuster [7] and Schütz et al. [11]. As far as possible, we clarified the information on heights and underlying rocks missing in the labels based on geological maps and descriptions of the areas in literature.

### 2.4. Mapping

The points on the maps are placed according to the coordinates available in the labels, or calculated from the description of the locations. The distribution maps of *A. megaspora* and *A. alpicola* are based on label data of the sequenced specimens, with the exception of eastern North America and the type locality of *A. alpicola* in western North America. For eastern North America, all published data and data represented in the GBIF, both as *A. endiviifolia* (*Pellia endiviifolia*) and *A.* (*Pellia*) *megaspora*, were referred by us to *A. megaspora* based on the conclusions of Schuster [7] and the sequenced topotype of the species. We do not provide the distribution map of *Apopellia endiviifolia*, since we do not yet have sufficient data to correct the map published by Schütz et al. [11], but we add the locality of one sequenced specimen of *A. endiviifolia* to the distribution map of *A. megaspora* and *A. alpicola* to support the occurrence of *A. endiviifolia* in the same area as the two other species.

### 2.5. Sampling for Molecular Analyses

The ITS1-2 nrDNA was first selected as the molecular marker of Pelliaceae and the *trn*L-F cpDNA was chosen, followed by the dataset from Schütz et al. [11] and our accumulated DNA markers database of liverworts. Both genomic regions were recommended as appropriate barcodes for distinguishing cryptic species in the related simple thalloid liverwort genus *Aneura* [12]. 36 specimens of Pelliaceae were selected for molecular analyses: 31 specimens from different regions of Russia, two specimens from Japan, two specimens from USA (incl. the topotype specimen *A. megaspora* from Massachusetts) and a single specimen from China (Table 1). The genus *Apopellia* was represented by 29 specimens and the genus *Pellia* by 7 specimens. Additionally, for producing datasets for phylogenetic estimation, previously published accessions were used: 61 ingroup and four outgroup specimens with *trn*L-F data from Schütz et al. [11], accessions of two ingroup and three outgroup specimens from Konstantinova et al. [13,14], *trn*L-F sequences from the chloroplast genome of seven *Apopellia* species from Grosche et al. [15] and Sawicki et al. [16] and the ITS1-2 sequence data for two specimens from GenBank.

**Table 1.** The list of specimens molecularly tested in current study with voucher details and GenBank accession numbers.

| Taxon | Specimen Voucher | GenBank Accession Number | |
| --- | --- | --- | --- |
| | | ITS1-2 nrDNA | *trn*L-F cpDNA |
| *Apopellia alpicola* (R.M. Schust. ex L. Söderstr., A. Hagborg & von Konrat) Nebel & D. Quandt | Russia: Murmansk Prov., N. Konstantinova, NK41-4-94, 1010 (KPABG) | OQ832113 | OQ817877 |
| | Russia: Trans-Baikal Terr., O. Afonina, OAf1706, 113953 (KPABG) | OQ832111 | OQ817875 |
| | Russia: Trans-Baikal Terr., Yu. Mamontov, 266-1-2568 (MHA) | OQ968335 | OQ957420 |
| | Russia: Tuva Rep., V. Bakalin, VB-99-9-1...10, 100916 (KPABG) | OQ832110 | OQ817874 |
| | USA: Alaska, N. Konstantinova, K73-92, 124324 (KPABG) | OQ832112 | OQ817876 |
| *A. endiviifolia* (Dicks.) Nebel & D. Quandt cryptic species A2 | Russia: Adygeia Rep., N. Konstantinova, K437-2-12, 115936 (KPABG) | OQ832120 | OQ817884 |
| | Russia: Adygeia Rep., N. Konstantinova, K437-1b-12, 115935 (KPABG) | OQ832118 | OQ817882 |
| | Russia: Karachaevo-Cherkessia Rep., N. Konstantinova, K128-1-20 (KPABG) | OQ832115 | OQ817879 |
| | Russia: Karachaevo-Cherkessia Rep., N. Konstantinova, K101-2-20, 123995 (KPABG) | OQ832116 | OQ817880 |

**Table 1.** *Cont.*

| Taxon | Specimen Voucher | GenBank Accession Number | |
| --- | --- | --- | --- |
| | | ITS1-2 nrDNA | *trn*L-F cpDNA |
| | Russia: Karachaevo-Cherkessia Rep., N. Konstantinova, K111-20 (KPABG) | OQ832119 | OQ817883 |
| | Russia: Krasnodar Terr., N. Konstantinova, K-460-1-12 (KPABG) | OQ832117 | OQ817881 |
| *A. endiviifolia* (Dicks.) Nebel & D. Quandt cryptic species EU | Russia: Altay Terr., 9085521 (MHA) | OQ968340 | OQ957425 |
| | Russia: Krasnodar Terr., N. Konstantinova, K-14-1-14 (KPABG) | OQ832121 | OQ817885 |
| | Russia: Krasnodar Terr., M. Kozhin, 01.06.2017 (KPABG) | OQ832122 | OQ817886 |
| *A. endiviifolia* (Dicks.) Nebel & D. Quandt cryptic species A1 | Russia: Primorsky Terr., E. Borovichev, BB-6-1d-14, 119443 (KPABG) | OQ832114 | OQ817878 |
| | Russia: Kamchatka Terr., V. Bakalin, K-57-9-21, 9085523 (MHA) | OQ968339 | OQ957424 |
| *A. megaspora* (R.M. Schust.) Nebel & D. Quandt | Russia: Altay Terr., 9085519 (MHA) | OQ968337 | OQ957422 |
| | Russia: Arkhangelsk Prov., 9085547 (MHA) | OQ968338 | OQ957423 |
| | Russia: Buryatia Rep., N. Konstantinova, 40-01, 102436 (KPABG) | OQ832102 | OQ817866 |
| | Russia: Irkutsk Prov., V. Bakalin, Irk-83-3-22 (VBGI) | OQ832104 | OQ817868 |
| | Russia: Komi Rep., M. Dulin, MVD-1104, 116783 (KPABG) | OQ832109 | OQ817873 |
| | Russia: Krasnoyarsk Terr., Taimyr, V. Fedosov, 13-3-1022, 116853 (KPABG) | OQ832100 | OQ817864 |
| | Russia: Krasnoyarsk Terr., Taimyr, V. Fedosov, 13-3-0420, 116907 (KPABG) | OQ968336 | OQ957421 |
| | Russia: Trans-Baikal Terr., Yu. Mamontov, 533-6-7183 (MHA),125752 (KPABG) | OQ832105 | OQ817869 |
| | Russia: Buryatia Rep., Yu. Mamontov, 574-1-5156 (MHA),125753 (KPABG) | OQ832106 | OQ817870 |
| | Russia: Buryatia Rep., Yu. Mamontov, 575-1-5085 (MHA), 125754 (KPABG) | OQ832107 | OQ817871 |
| | Russia: Buryatia Rep., Yu. Mamontov, 601-6-5046 (MHA), 125755 (KPABG) | OQ832108 | OQ817872 |
| | Russia: Tuva Rep., T. Otnyukova, G100790 (KPABG) | OQ832101 | OQ817865 |
| | USA: Massachusets, N. Konstantinova & R. Schuster, AMass14-92, 123492 (KPABG) topotype | OQ832103 | OQ817867 |
| *P. epiphylla* (L.) Corda | Russia: Murmansk Prov., N. Konstantinova, K72-1-05, 14114 (KPABG) | OQ832126 | OQ817890 |
| *P. neesiana* (Gottsche) Limpr. | Russia: Kamchatka Terr., V. Bakalin, K-50-33-02-VB, 104126 (KPABG) | OQ832123 | OQ817887 |
| | Russia: Murmansk Prov., E. Borovichev, BE5-4-09, 19070 (KPABG) | OQ968341 | OQ957426 |
| | Japan: Honsu, T. Yamaguchi, 524, 118849 (KPABG) | OQ832124 | OQ817888 |

**Table 1.** *Cont.*

| Taxon | Specimen Voucher | GenBank Accession Number | |
|---|---|---|---|
| | | ITS1-2 nrDNA | *trn*L-F cpDNA |
| | Japan: Honsu, T. Yamaguchi, Bryophytes of Asia, Fasc. 26 (2019), No. 644, 124727 (KPABG) | OQ832125 | OQ817889 |
| *Pellia* sp. indet. | China: Yunnan Prov., Gongshan Country, D. Long & J. Shevock, 37188 (KUN) | OQ832128 | OQ817892 |
| *Pellia* sp. nov. Konstant., Mamontov, Vilnet | Russia: Sakhalin Prov., Kuril Isl., Shumshu Isl., V. Bakalin, K-125-28-04, 107679 (KPABG) | OQ832127 | OQ817891 |

*2.6. DNA Isolation, PCR Amplification and DNA Sequencing*

DNA from apical parts of dried thalli carefully cleaned from substrate was extracted with DNeasy Plant Mini Kit (Qiagen, Germany) according to the manufacturer's protocol. The ITS1-2 and *trn*L-F sequence data were amplified and sequenced with pairs of primers described in White et al. [17] and Taberlet et al. [18]. PCR was carried out in 20 µL volumes with the following amplification cycles: 3 min at 94 °C, 30 cycles (30 s 94 °C, 40 s 56 °C, 60 s 72 °C) and 2 min of final extension time at 72 °C. Amplified fragments were visualized on 1% agarose TAE gels by EthBr staining, purified using the Cleanup Mini Kit (Evrogen, Moscow, Russia), and used as a template in sequencing reactions with the ABI Prism BigDye Terminator Cycle Sequencing Ready Reaction Kit (Applied Biosystems, Waltham, MA, USA) following the standard protocol provided for 3100 Avant Genetic Analyzer (Applied Biosystems, USA).

*2.7. Phylogenetic Analysis*

The nucleotide sequence data obtained here were assembled with the program BioEdit 7.0.1 [19]. Pelliaceae is characterized by a number of extended indels during intron and spacer regions of *trn*L-F. To exclude ambiguities in the alignment of indels and achieve a congruent result of phylogenetic estimation, the alignment of the *trn*L-F region was downloaded from Schütz et al. [11] and newly generated sequences were incorporated into it. The ITS1-2 dataset was automatically aligned with the ClustalW option and then manually corrected in BioEdit 7.0.1. All positions were taken into account, and absent data were coded as missing. The preliminary test of phylogenetic congruence of both datasets reveals different affinity in the genus *Apopellia*, although it was insufficiently supported. Thus, phylogenetic estimations were conducted separately for each dataset.

The maximum likelihood analysis (ML) was performed with IQ-TREE [20], and the Bayesian analysis (BA) with MrBayes v. 3.2.1 [21]. The ModelFinder [22] resolved the HKY + I model as the best fit evolutionary model of nucleotide substitutions for the *trn*L-F dataset, and TIM + I + G—for the ITS1-2 dataset. Recommended models of nucleotide substitutions, with four rate categories of gamma distribution to evaluate the rate of heterogeneity among sites in case of ITS1-2 dataset, and ultrafast bootstrapping [23] with 1000 replicates were used as parameters for ML analyses. The obtained ML tree topologies were redrawn in NJplot [24]. For the Bayesian analysis, both datasets were assigned the GTR + I + G model as recommended by the program's creators; gamma distributions were approximated with four rate categories. Two independent runs of the Metropolis-coupled MCMC were used to sample parameter values in proportion to their posterior probability. Each run included three heated chains and one unheated chain, and the two starting trees were chosen randomly. The number of generations was one million for the ITS1-2 dataset and two million for the *trn*L-F dataset. Trees were saved every 100th generation. The average standard deviation of split frequencies between two runs in the ITS1-2 analysis was 0.008176 and 0.008953 in *trn*L-F. For the ITS1-2 calculations the 2500 (25%) trees were discarded in each run, and 15,000 trees from both runs were sampled after burning, while for *trn*L-F 5000 (25%) trees were discarded in each run, and 30,000 trees from both runs

were sampled after burning. Bayesian posterior probabilities were calculated from trees sampled after burn-in. The FigTree v.1.3.1 was used to reconstruct the phylogeny after Bayesian estimation [25].

The infraspecific and interspecific sequence variability was estimated as the average pairwise *p*-distances for ITS1-2 and *trn*L-F in Mega 11 [26] using the pairwise deletion option for counting gaps.

## 3. Results

### 3.1. DNA Estimation

The ITS1-2 and *trn*L-F sequence data were obtained for all 36 molecularly tested specimens of Pelliaceae and deposited into GenBank (Table 1). In total, the ITS1-2 dataset includes 43 accessions and the *trn*L-F dataset 109 accessions. The ML calculation of the ITS1-2 dataset resulted in a single tree with the arithmetic mean of Log likelihood −6451.960, while in the BA analysis means of Log likelihood for both sampled runs were −6504.16 and −6501.37, respectively. Tree topologies achieved from both estimations were identical, and thus Figure 1 demonstrates a ML tree achieved by an ITS1-2 dataset with indication of bootstrap support (BS) values from ML analyses and Bayesian posterior probabilities (PP) from BA. The arithmetic mean of Log likelihood in the ML calculation of the *trn*L-F dataset was −2195.220, while in the BA analysis means of Log likelihood for both sampled runs were −2327.27 and −2326.93. The differences between obtained *trn*L-F topologies consist in the position of the specimen *Pellia* sp. K-125-28-04 from Shumshu I., Russia: and basal for *P. neesiana* + *P. appalachiana* + *Pelli*a sp. 37188 from Yunnan Province, China without support in ML tree or basal to *Pellia*-clade as a whole, with support PP = 1.00 in BA. Figure 2 demonstrates an ML tree of a *trn*L-F dataset with the indication of bootstrap support (BS) values from ML analyses and Bayesian posterior probabilities (PP) from BA, and the position of *Pellia* sp. K-125-28-04 in BA marked by a dotted line.

The backbone topologies obtained here from ITS1-2 and *trn*L-F resemble those published in Schütz et al. [11] based on three cpDNA loci (*trn*L-F, *rps*4, *rpl*16). From nuclear and chloroplast markers, the family Pelliaceae splits into two main clades that are corresponded to the genera *Pellia* (BS = 96%, PP = 1.00 or 96/1.00 in ITS1-2, 60/1.00 in *trn*L-F) and *Apopellia* (98/1.00 in ITS1-2, 99/1.00 in *trn*L-F). In the *trn*L-F topology, the clade of the genus *Pellia* combines related sister clades of *P. epiphylla* and *P. neesiana* with subsequently related phyla of *P. appalachiana* (59/0.92) and specimen of *Pellia* sp. 37188 from China (64/0.99). The subsequently diverged position of specimen *Pellia* sp. K-125-28-04 from Shumshu I. is unsupported in ML, and robustly supported in BA (1.00) as a basal of the *Pellia*-clade. The single specimen of *P. epiphylla* from Murmansk Province was placed with that previously sequenced by Schütz et al. (2016) (72/0.93). Within *P. neesiana*-clade (68/0.99), three subclades could be revealed: a specimen from Kamchatka Territory clustered with several USA and European specimens (87/-); a specimen from the Republic of North Ossetia-Alanya and Murmansk Province clustered with European ones (82/0.58); and two specimens from Honshu I. clustered with a previously published Japanese accession (87/0.95). In the ITS1-2 topology, distribution of *Pellia* specimens among clades and their relation agreed with results from *trn*L-F. Within *Apopellia*, six clades could be recognized from *trn*L-F and five from ITS1-2 that corresponded with the topology of Schütz et al. [11], though *A. endiviifolia* appears not to be monophyletic from both markers, in contrast to ambiguously supported monophyly achieved in Schütz et al. [11]. The A1 subclade of *A. endiviifolia* extended by our single specimen from Primorsky Territory has been clustered with accession from South Korea without support, and comprised a separate linage from four Japanese and single Kamchatka's accessions (84/0.77) of clade A1. The subclades of *A. endiviifolia* marked as EU and A2 in Schütz et al. [1] are kept here, while the EU clade is complemented by two specimens from Krasnodar Territory (Caucasus), one accession from Republic of Altai and four accessions from chloroplast genomic studies from Germany, Poland and the Czech Republic, and the A2 (95/1.00) subclade is complemented by six accessions from Caucasus and three accessions from chloroplast genomic studies from Poland and the Czech Republic.

Three subclades of *A. endiviifolia* could also be recognized in ITS1-2 topology, with specimens sequenced here. The specimen composition of the subclades EU, A2 and A1 fully agrees with those shown in the *trn*L-F analysis. Eighteen specimens sampled from different regions of Russia, as well as from Alaska and Massachusetts (topotype), dispersed in two sister related clades of *A. megaspora* (87/0.80) were marked by Schütz et al. [11] as E and W. Thirteen accessions (including those from the topotype) are combined with three accessions from Canada (Alberta) and one from the USA (Michigan); following Schütz et al. [11], this clade E (100/1.00) kept its name here as *A. megaspora*. This clade is subdivided into three subclades. The first one includes three accessions from Alberta and Taimyr Peninsula (98/0.93), the second combines ten accessions from the Asiatic part of Russia (96/1.00) and the third subclade consists of two accessions from USA, including the topotype (90/-). The specimen 13-3-0420 from Taimyr Peninsula was placed in an unresolved relation of this clade.

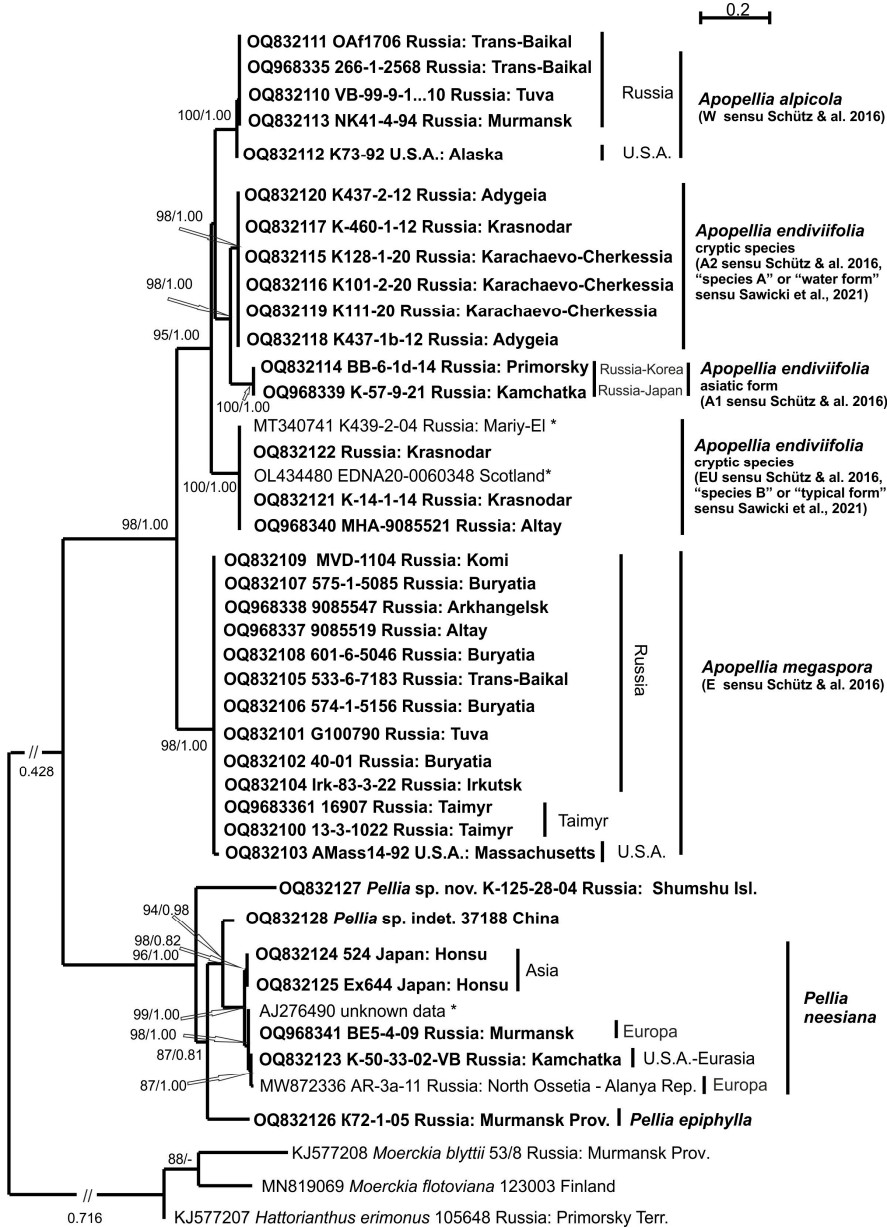

**Figure 1.** The ML phylogram for the family Pelliaceae reconstructed from ITS1-2 nrDNA of 36 accessions. Bootstrap supports from maximum likelihood and Bayesian posterior probabilities of more than 50% (0.50) are indicated. The accessions downloaded from GenBank and included only in ITS1-2 nrDNA analyses are marked by an asterisk, and newly obtained accessions are in bold.

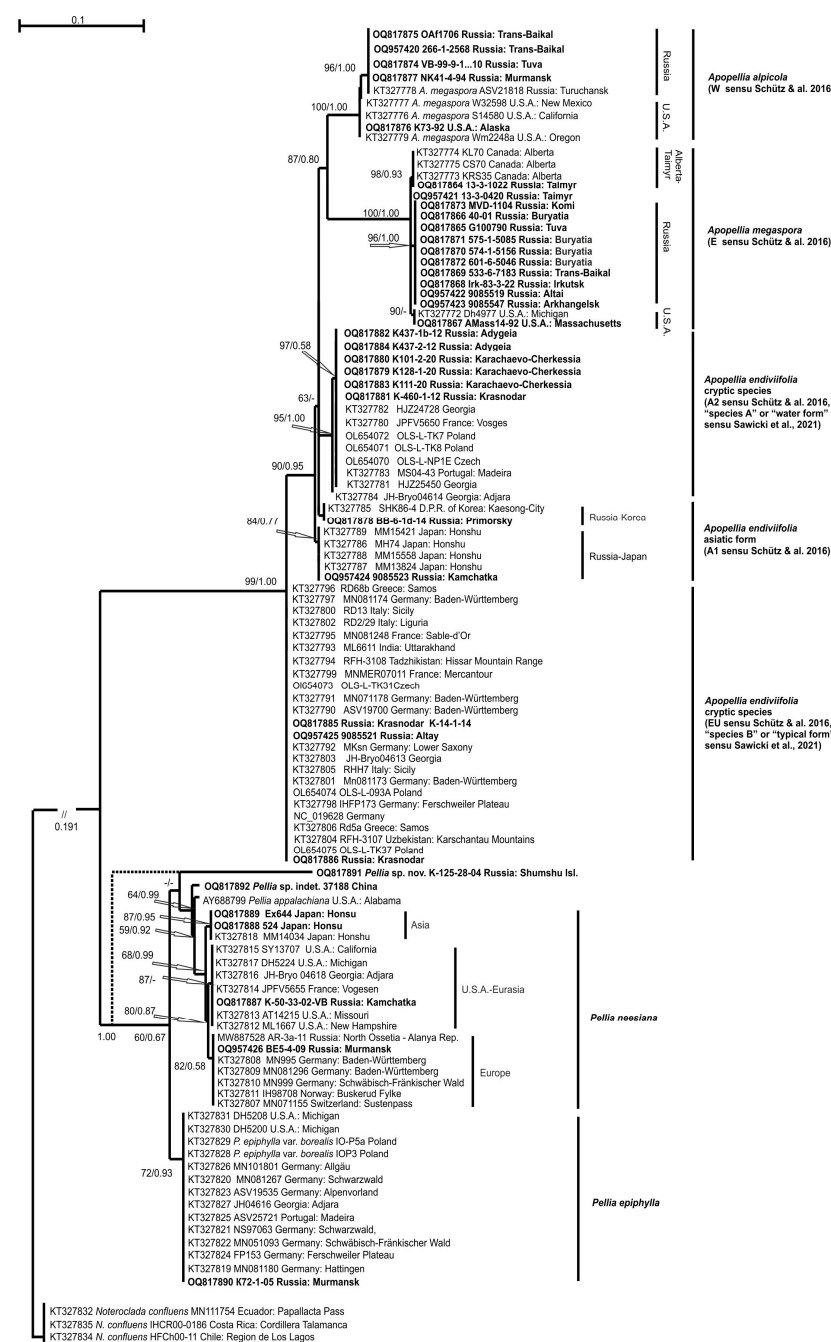

**Figure 2.** The ML phylogram for the family Pelliaceae reconstructed from *trn*L-F cpDNA of 102 accessions. Bootstrap supports from maximum likelihood and Bayesian posterior probabilities of more than 50% (0.50) are indicated. The position of a *Pellia* sp. nov. is marked by a dotted line in the BA phylogram. The newly obtained accessions are in bold.

The second clade (100/1.00), which is W according to Schütz et al. [11], includes four newly sequenced accessions from the European and Asian parts of Russia and a single one from Alaska, and is located subsequently with single Russian accessions (96/1.00) and three USA accessions. We provisionally call the specimens from this clade *A. alpicola* based on its distribution (but see discussion). This is based on ITS1-2, *A. alpicola*-clade placed within *A. endiviifolia*-clade but without support, whereas *A. megaspora* became a sister related to them (98/1.00). The ITS1-2 marker also reveals divergence between American and Russian specimens in both *A. megaspora* and *A. alpicola*.

The estimate of infra- and interspecies genetic distances (*p*-distances) was provided for each taxon-clade, including subclades (Table 2). Within subclades, the *trn*L-F variation is mostly absent, with exception of the clade *A. alpicola* USA (0.2%), the subclade of *P. neesiana* with Asiatic accessions (0.1%) and the *P. epiphylla*-clade (0.1%). The ITS1-2 became more variable within subclades: *A. alpicola* Russian accessions (0.1%), *A. endiviifolia* A2 (0.5%), *A. endiviifolia* EU (0.2%) and *P. neesiana* Europe (1.1%). The infraspecific variation within *A. alpicola* is 0.8% in *trn*L-F and 0.7% in ITS1-2 (or 0.8/0.7), within *A. megaspora* it is 0.2–0.4/0.5–1.6%, among all subclades of *A. endiviifolia* it is 0.5–3.0/6.0–12.7% and within *P. neesiana* it is 0.2–0.5/1.7–2.2%. *Apopellia alpicola* and *A. megaspora* differed by 7.7–8.1/18.3–20.1%, and *P. epiphylla* and *P. neesiana* by 3.6–4.1/16.6–18.7.

The level of variability in *A. endiviifolia* is greater than in other species, which additionally suggests its complex structure and possibly cryptic speciation. The Russian-Korean and Russian-Japanese subclades diverged in 0.5% in *trn*L-F and 1.1% in ITS2 (only the last locus was obtained for Russian accessions), which corresponded with levels of infraspecific variation in *A. alpicola* and *A. megaspora*. The separation of both Asiatic subclades of *A. endiviifolia* from *A. endiviifolia* A2 is 1.2–1.3/6.0–6.5, while from *A. endiviifolia* EU it is 1.7–1.8/10.5–12.7. The differences between *A. endiviifolia* subclades A2 and EU is 3.0/11.7, which is below the level of differentiation between the species pairs *A. alpicola–A. megaspora* and *Pellia epiphylla–P. neesiana*. *Apopellia alpicola* differs from all clades of *A. endiviifolia* by 1.8–4.8/8.6–12.6, and the most similarity is found with *A. endiviifolia* A1, with specimens from Russia and Korea. *Apopellia megaspora* is differentiated from all clades of *A. endiviifolia* (5.0–7.6/16.0–21.6) almost twice as high as *A. alpicola*. *Pellia appalachiana* differs from *P. neesiana* by 1.5–1.6% in *trn*L-F, and by 1.3% from the specimen *37188* of *Pellia* sp. from China. The specimen of *Pellia* sp. K-125-28-04 from Shumshu I. differs from other members of the genus *Pellia* by 9.4–9.9/21.3–24.4, which clearly suggested it as a previously unknown species that will be described in a separate paper.

### 3.2. Morphology

For the correct identification of the species of *Apopellia*, sporophytes or at least pseudoperianths are necessary. The results of the revision of the specimens of *Apopellia endiviifolia* (stored as *Pellia endiviifolia*) in KPABG showed that, for example, out of 100 specimens of *A. endiviifolia*, only five were with sporophytes, ca. 20 with pseudoperianths and ca. 30 with autumnal proliferations, while many specimens were either sterile or male, and then according to Schuster [5]: "useless" or "taxonomically meaningless". As a result, a large number of specimens in herbaria are incorrectly identified. On the other hand, young pseudoperianths are not always clearly visible, and are rarely abundant in the herbarium specimens. For example, in the specimens from the topotype, we managed to find several pseudoperianths only after repeated careful study of all the specimens (total of four specimens). However, some features of male and sterile plants can also be used; they and other morphological features are discussed in the taxonomical treatment section.

### 3.3. Distribution

The obtained results radically change the ideas about the distribution of the genus *Apopellia*. Firstly, we discovered *A. megaspora* (considered eastern American) in the south and north of East Siberia, the European Urals and the north-east of European Russia (Arkhangelsk Province), and confirmed its reports from western North America (Figure 3). At the same time, we attributed the previous record of *A. megaspora* from Yenisei River Basin, East Siberia [11] to *A. alpicola*. The latter species also turned out to be almost circumpolar, and was also found (in addition to western North America, from where it was described) in Europe (Murmansk Region) and Siberia both south and east (see specimens examined (Figure 3). *A. endiviifolia* (a Eurasian species, which is most widely distributed in Europe) is probably not rare in the Russian Far East, and is known from scattered localities in the mountains of south Siberia, Japan, South Korea, Central Asia (Uzbekistan, Tadzhikistan) and Himalaya (India).

**Table 2.** The values of *p*-distances (%) for genera *Apopellia* and *Pellia*, calculated from *trn*L-F and ITS1-2.

| | Taxon (with Subclade Indication According with Figures 1 and 2) | Variability within Subclade, *trn*L-F/ITS1-2, % | Variability among Subclades, *trn*L-F/ITS1-2, % | | | | | | | | | | | | | | |
|---|---|---|---|---|---|---|---|---|---|---|---|---|---|---|---|---|---|---|
| | | | 1 | 2 | 3 | 4 | 5 | 6 | 7 | 8 | 9 | 10 | 11 | 12 | 13 | 14 | 15 | 16 |
| 1 | *A. alpicola* Russia | 0.0/0.1 | | | | | | | | | | | | | | | | |
| 2 | *A. alpicola* USA | 0.2/n/c | 0.8/0.7 | | | | | | | | | | | | | | | |
| 3 | *A. megaspora* Alberta-Taymir | 0.0/n/c | 8.1/20.1 | 7.8/19.8 | | | | | | | | | | | | | | |
| 4 | *A. megaspora* Russia | 0.0/0.2 | 7.9/18.5 | 7.7/18.5 | 0.4/0.5 | | | | | | | | | | | | | |
| 5 | *A. megaspora* USA | 0.0/n/c | 8.1/18.3 | 7.8/18.3 | 0.3/1.4 | 0.2/1.6 | | | | | | | | | | | | |
| 6 | *A. endiviifolia* A2 | 0.0/0.5 | 3.8/11.8 | 3.0/11.8 | 6.3/21.6 | 6.4/20.5 | 6.5/20.3 | | | | | | | | | | | |
| 7 | *A. endiviifolia* A1 Russia-Korea | 0.0/n/c | 2.6/8.8 | 1.8/8.6 | 5.0/16.9 | 5.1/16.0 | 5.3/16.6 | 1.2/6.5 | | | | | | | | | | |
| 8 | *A. endiviifolia* A1 Russia-Japan | 0.0/n/c | 3.0/12.6 | 2.2/12.3 | 5.2/20.6 | 5.4/19.1 | 5.5/19.2 | 1.3/6.0 | 0.5/1.1 | | | | | | | | | |
| 9 | *A. endiviifolia* EU | 0.0/0.2 | 4.8/10.5 | 4.5/10.3 | 7.5/19.7 | 7.6/18.2 | 7.6/18.2 | 3.0/11.7 | 1.8/10.5 | 1.7/12.7 | | | | | | | | |
| 10 | *Pellia* sp. nov. Russia | n/c/n/c | 19.0/38.3 | 18.7/38.6 | 17.9/39.1 | 18.3/35.6 | 17.9/36.1 | 14.2/39.7 | 13.3/27.7 | 13.3/33.8 | 16.8/34.7 | | | | | | | |
| 11 | *Pellia* sp. indet. China | n/c/n/c | 16.9/24.0 | 16.3/24.3 | 16.7/25.1 | 17.6/22.3 | 16.8/23.9 | 12.6/27.5 | 11.7/24.9 | 11.7/24.6 | 14.3/23.4 | 9.5/21.3 | | | | | | |
| 12 | *P. appalachiana* | n/c/- | 16.9/- | 16.3/- | 16.2/- | 17.6/- | 16.6/- | 12.4/- | 11.6/- | 11.2/- | 13.8/- | 9.7/- | 1.3/- | | | | | |
| 13 | *P. neesiana* Asia | 0.1/0.0 | 16.9/35.2 | 16.3/35.0 | 16.6/34.7 | 17.6/33.3 | 16.7/33.5 | 12.8/37.0 | 12.0/28.8 | 11.9/32.9 | 13.6/32.9 | 9.9/23.6 | 2.1/7.6 | 1.6/- | | | | |
| 14 | *P. neesiana* USA-Eurasia | 0.0/n/c | 17.1/40.7 | 16.5/40.6 | 16.6/42.7 | 17.7/38.9 | 16.9/39.0 | 13.0/41.8 | 12.2/26.6 | 11.9/34.8 | 13.7/35.6 | 9.4/23.0 | 2.0/8.2 | 1.5/- | 0.5/2.2 | | | |
| 15 | *P. neesiana* Europe | 0.0/1.1 | 16.7/42.2 | 16.1/42.2 | 16.2/43.3 | 17.3/40.5 | 16.5/40.5 | 12.6/42.8 | 11.8/28.3 | 11.5/37.0 | 13.3/37.1 | 9.7/24.4 | 2.0/9.1 | 1.5/- | 0.5/2.2 | 0.2/1.7 | | |
| 16 | *P. epiphylla* | 0.1/n/c | 16.9/38.7 | 16.0/38.9 | 15.9/41.2 | 17.1/37.8 | 16.1/37.7 | 13.3/40.4 | 12.2/23.5 | 11.9/32.7 | 13.9/34.3 | 9.5/24.3 | 3.2/12.2 | 3.4/- | 4.1/16.6 | 3.8/17.8 | 3.6/18.7 | |

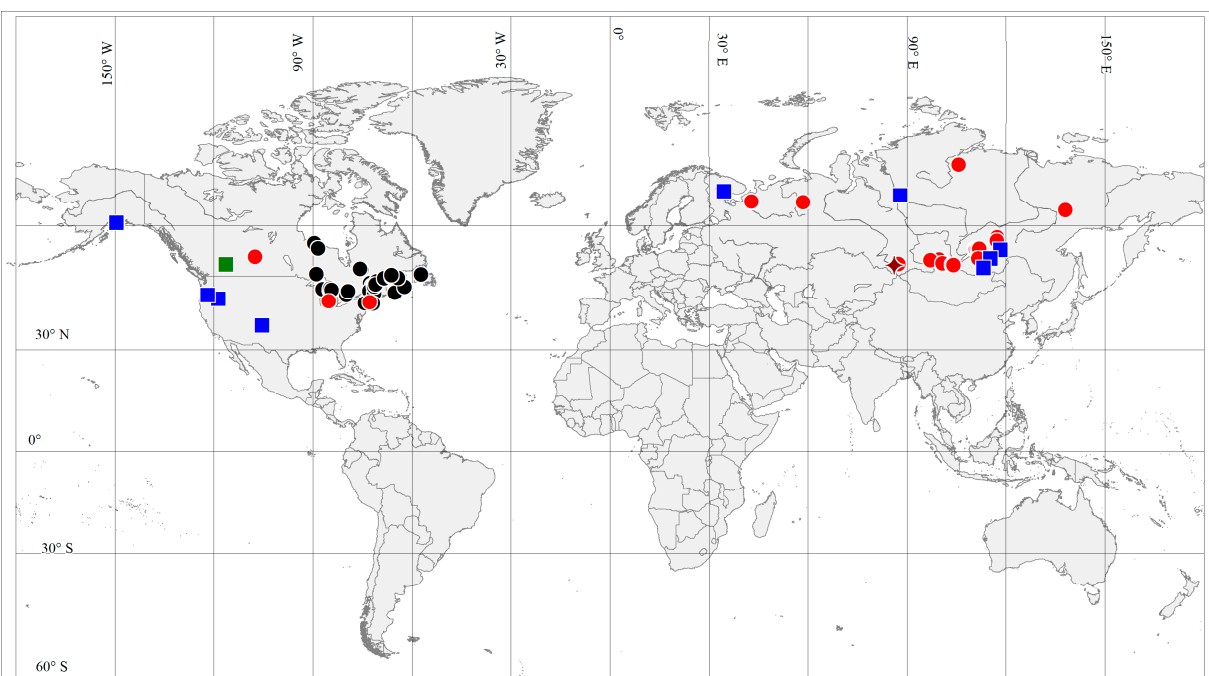

**Figure 3.** Distribution of *Apopellia megaspora* (red dots represent localities of sequenced specimens, black dots localities based on Schuster, 1992) and *A. alpicola* (blue squares represent localities of sequenced specimens, green square locality based on Schuster, 1981); brown rhomb sequenced specimen of *Apopellia endiviifolia*.

## 4. Discussion

Morphologically, the genus *Apopellia* differs from *Pellia* s.str. quite well, especially when pseudoperianths or extremely characteristic autumn proliferations are developed. However, differentiation of sterile thalli needs careful microscopic study, such as the shape of the hairs on the ventral side of the thalli and study of the cross-section. In addition, these features are not always clearly visible, and the shape, distribution and number of cells in slime hairs, especially in old herbarium specimens, are sometimes very difficult to see. As a consequence, misidentifications of *Pellia* as *Apopellia* and vice versa are not rare in herbaria.

The first results of phylogenetic analyses inferred from tRNALeu (CAA) gene spacer sequences were doubtful [27], but recently implemented molecular phylogenetic study of significantly sampled Pelliaceae based on three cpDNA markers robustly supports divergence of *Apopellia*, which has then started to be treated as a distinct genus [11]. The phylogeny of Pelliaceae obtained here from ITS1-2 nrDNA resulted in two clades composed by species *Pellia* and *Apopellia*, which deeply diverge from each other (*p*-distances 22.0–43.6%, Table 2). Thus, we found additional support from nuclear DNA for the treatment of *Apopellia* as a separate genus.

Expanded sampling in our study showed that the subclades interpreted by Schütz et al. [11] as "*A. megaspora* W" and "*A. megaspora* E" kept their separation from both chloroplast and nuclear DNA markers. The topotype of *A. megaspora*, nested within the clade E, makes it possible to confidently attribute this clade to *A. megaspora*. At the same time, a morphological study of several newly sequenced specimens from Russia and Alaska located in the clade called W in Schütz et al. [11] showed that they matched the description of *A. alpicola* by Schuster [6]; thus, we attribute all specimens from this clade to *A. alpicola*. Notably, all sequenced specimens from Alberta are located in *A. megaspora* clade. At first glance, this is unclear, since *A. alpicola* was described from Alberta [6] too. However, in South Siberia and in the south of East Siberia, both taxa also occur at fairly close distances (less than 50 km in some places), which is confirmed by sequencing data, unlike Alberta where sequences data were obtained for *A. megaspora* only. Both species occur at more or

less similar elevation of about 1000 m altitude, specifically, in the Republic of Tuva in Siberia and in the mountains of Alberta in North America. Thus, there is a sympatric distribution of two species due to environmental preferences, specifically *A. megaspora* restricted to Ca-rich habitats, whereas *A. alpicola* occurs in acidic or neutral habitats.

The data obtained show that *A. megaspora* has disjunctive, almost circumpolar distribution (Figure 3), but it is most common in eastern North America, while *A. alpicola* occurs scattered in Eurasia and seems widespread in western North America, but not yet found in eastern North America. *A. endiviifolia* is most common in Europe, while east of Eurasia occurring scattered in mountains of South Siberia and Central Asia, and it has not yet been found in North America. We also found some genetic differences of western, eastern American and Eurasian specimens of *A. alpicola* and *A. megaspora* (Figures 1 and 2), which indicates a fairly long-standing divergence and isolation of populations of these two species in these areas.

The very variable *Apopellia endiviifolia* is most common in temperate regions of Europe, which is probably why it is the best studied. In a study of *A. endiviifolia* infraspecific structure about forty years ago, two cryptic species were detected by isoenzyme markers and named as typical (species B) and aquatic (species A) forms [28]; later, these two forms were confirmed with DNA markers [29], and recently from chloroplast genome data [16]. Schütz et al. [11] suggested monophyly of *A. endiviifolia* with three genotypes (EU, A2, A1), and treated them as patterns of geographical distribution and ecological forms. The ITS1-2 and *trn*L-F tests of additional samples of *A. endiviifolia* from Russia, with special attention to their ecology, support evidence that *A. endiviifolia* presents a complex of cryptic taxa with sympatric distribution. Asian accessions could also be a cryptic species [9,28], and possibly more than one based on paraphyly achieved by *trn*L-F. Nucleotide sequence evidence obtained in our study supports all previously distinguished genetic forms of *A. endiviifolia* in Russia: typical (EU); aquatic (water, A2); and Asiatic (Japanese, A1).

In spite of some progress, it is obvious that we still are far from finding out the actual distribution of the species of the genus. The problem is that sterile or male plants of the three species are almost indistinguishable, and sporophytes are infrequent. They are formed in spring, and depending on the region, are presented in specimens collected in May and early June. In *A. alpicola*, sporophytes have not yet been described at all. The revision of the specimens in KPABG showed that only about one third of the *Apopellia* specimens stored in the herbarium have perianth or sporophytes, and so can be accurately identified. Therefore, to find out the true distribution of species of the genus, it is essential to purposefully collect *Apopellia* species with sporophytes or perianths and carefully revise the collections of herbaria based on the detailed description below.

The inclusion of additional specimens of *P. neesiana* from Japan allows us to separate the distinct subclade with Asiatic accession, as well as two other subclades that partially correspond with USA-Eurasian and European distribution. The level of infraspecific *p*-distances is similar to infraspecific variation of *A. megaspora*, and evidently reflects an internal species structure that could not be taxonomically identified.

One of the unexpected results was that one specimen from Shumshu Isl. (Figures 1 and 2), identified as *Pellia endiviifolia*, differs genetically and morphologically from the known species of *Pellia* and *Apopellia*. This specimen will be described in detail in a separate paper.

## 5. Taxonomic Treatment

*Apopellia alpicola* (R.M. Schust. ex L. Söderstr., A. Hagborg & von Konrat) Nebel & D. Quandt [*Pellia alpicola* R.M. Schust. ex L. Söderstr., A. Hagborg & von Konrat in Phytotaxa 76(3): 39. 2013 ≡ "*Pellia endiviifolia* subsp. *alpicola*" R.M. Schust. in J. Hattori Bot. Lab. 70: 145. 1991, nom. inval. (Art. 40.7; herbarium not specified) – Holotype: Canada, Alberta, Helen Lake, Banff Natl. Park, *Schuster 85-721* (F n.v.)] (Figure 4).

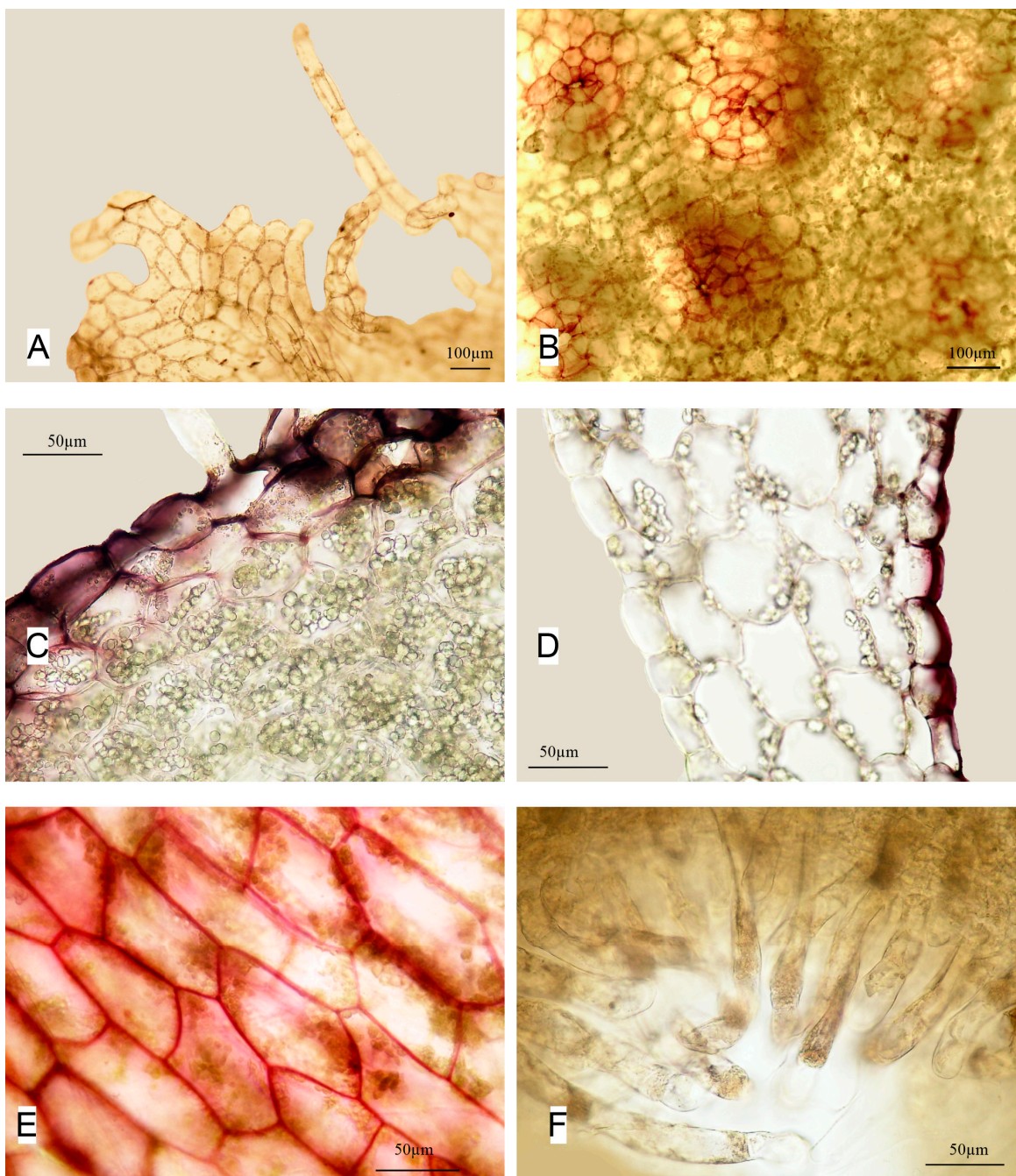

**Figure 4.** *Apopellia alpicola*: (**A**)—perianth mouth; (**B**)—groups of antheridia cavities; (**C**)—cross-section of middle part of thallus with oil-bodies in the cells; (**D**)—cross-section, transition part to the "wing"; (**E**)—cells of dorsal side of thallus; and (**F**)—slime hairs near apex (all from NK41-4-94, KPABG).

Plant lamina with wings and area near apex grass green contrasting with massive brown, dark red-brown to almost black costa, that on ventral side often is dark purple to red-purple, in the areas adjacent to the costa sometimes also purple, dark red or even light violet and dark red-brown to blood red colored (the color is distinct on the ventral side, especially when viewed under a light microscope; dried specimens in herbarium look dark brown to red brown in the costal area and green in wings), below the whole lamina almost black, (3–)5–9 mm wide and to 2–several cm long. Thalli are in several rows along margins one cell thick, then gradually thicken and turn into the very wide costa that is up to 1–2 mm (to ca. 30% of thallus) wide in cross section 7–12 cells thick, very gradually turning

into a 5–3 layer thallus and one-celled wings; cells of epidermis on the ventral side with purple to red brown walls (22–)25–35 × 25–40 μm, in the middle very large, thin walled, (30–)35–50 × (35–)40–75 μm, each a part of cells of epidermis with numerous oil bodies, that are colorless, spherical to ellipsoidal from one to three segments, ca. 5–7 × 5–9 μm. Cells of the dorsal side of the thin-walled thallus are very variable, along the margin in several rows from distinctly elongated along the margin and (30–)37(–50) × 40–75 μm. to almost isodiametric ca. 45–50 μm, then mostly obliquely directed to the costa. Slime hairs are numerous on the ventral side near the apex, and consist mostly of 3–5 cells ca. (20–)25–32 × 45–60(75–100) μm and scattered on sides of lamina or far below, mostly near costa, light brown or colorless and than hardly visible, consisting of 3–6 distinctly elongated cells ca. 25–30 × 50–100 μm. Rhizoids golden-yellow to golden-bronze are numerous, forming dense mats in mature plants and scattered, but very long in young, almost translucent plants. Vegetative propagation is not known.

Dioicous. Antheridia are in large groups of up to 30 tubercules, each with very large sessile antheridium ca. 350–400 μm in diameter. Perichaetium are flattened with peculiar wing-like outgrowths, short anteriorly and much longer posteriorly, resembling that of *Pellia neesiana*, but with very strongly lobed ciliated, and at the same time the short anteriorly part with long cilia, while on the posterior part cilia are shorter and lobes are weakly pronounced. One seriate cilia is up to several elongated cells that are (35–)40–60 × 75–150 μm. Sporophytes are not known.

Selected specimens examined. **USA**: Alaska, Kenai Mts., 60.5° N, 149.5° W, on peat soil in *Alnus* and *Prunus* thickets, 01.07.1992, Konstantinova K73-92 [KPABG 124324]. **RUSSIA:** Murmansk Province, Kandalaksha State Nature Reserve, Turiy Peninsula, on peat along bank of Khyam Brook, 14.08.1994, Konstantinova & Schuster, NK41-4-94 (per.); Republic of Tuva, Todzha Valley, valley of Azas Lake, on peat bank of brook, 13.07.1999, V.A. Bakalin VB-99-9-1....10 [KPABG 100916]; Buryatiya Republic, South Muya Range, Dzhirga River basin, 54°54′34.1″ N. 111°18′46.1″ E, 604 m a.s.l., *Alnus* sp.-*Betula* sp. grass boggy forest, on soil of bank of dry channel, 08.08.2013 Mamontov 396-2-1 [MHA-9085553], with *Chiloscyphus polyanthos, Plagiochila porelloides*; Trans-Baikal Territory, Alkhanai National Park, 50°54′00 N, 113°09′00″ W, rocks on bank of Ilya River, 8.07.2011, O.M. Afonina OAF1706 [KPABG 113953], with *Marchantia polymorpha* subsp. *ruderalis, Plagiochila porelloides, Chiloscyphus* sp.; Daurskiy Range, 52°57′26.0″ N. 115°16′31.8″ E. 639 m a.s.l., *Larix dahurica-Betula* sp. forest, on submerged ground on the shore of a rivulet, 14.07.2012, Mamontov 266-2[MHA-9085556], with *Plagiochila porelloides, Marchantia polymorpha* subsp. *ruderalis.*

Ecology. Mountain species occurring on damp peat banks of brooks and lakes, on silt-covered rocks along edges and in temporarily drying up stream beds. In contrast to *A. megaspora*, which is a strictly calciphilous species, *A. alpicola* is found on neutral or slightly acidic sites with e.g., *Marchantia polymorpha* subsp. *ruderalis, Plagiochila porelloides, Chiloscyphus* sp.

Schuster [7] postulated that *Apopellia* (*P. alpina*) occurs in western North America "along or in alpine rills, where under acidic conditions".

Distribution. The species can be characterized as mostly sub-arctomontane and almost circumpolar, but its distribution is poorly known because of misidentifications as *A. endiviifolia.* The species occurs at relatively high elevations or high latitudes. Sequenced specimens of species from the west of North America were collected in California at 43°30′ N at elevation of ca. 2682 m (8800 ft) and in New Mexico (ca. 36°36′ N) at more than 3048 m (10,000 ft). The species was found in south Siberia and in Alaska at heights of about 600–900 m alt. The species was collected almost near sea level in the northernmost locality in Murmansk Province (beyond the Arctic Circle).

*Apopellia endiviifolia* (Dicks.) Nebel & D.Quandt ≡ *Jungermannia endiviifolia* Dicks., Fasc. Pl. Crypt. Brit. 4: 19. 1801 ≡ *Pellia endiviifolia* (Dicks.) Dumort., Recueil Observ. Jungerm.: 27. 1835—Type: Great Britain, Scotland, in sylvis humidis Scotiae, *A. Menzies s.n.* (E, barcode E00007414) (Figure 5).

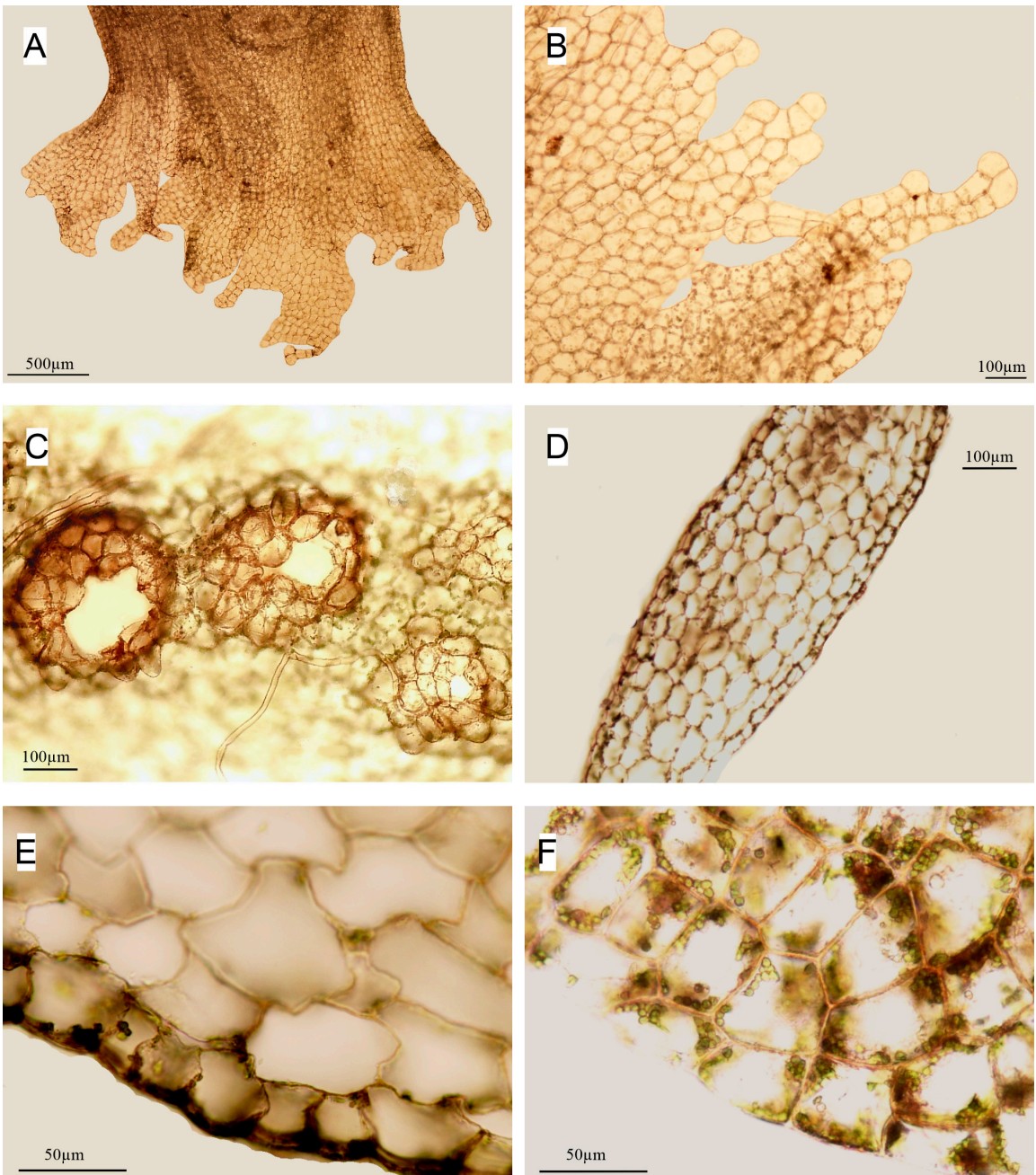

**Figure 5.** *Apopellia endiviifolia* (**A,B**)—perianth mouth; (**C**)—antheridia cavities; (**D**)—cross-section; (**E**)—part of cross-section showing small cells of epidermis contrasting with large cells of medula; (**F**)—cells of dorsal side of thallus along margin; (all from KPABG 113353).

Plants are from light to dark green sometimes with a dark red brown-colored central part and base of pseudoperianth, and in herbaria often dark green to blackish and blackish-green, sometimes with smell. Rhizoids are light brown to bronze-brown and red-brown, absent near apices and in linear water forms but numerous in terrestrial plants, on the ventral surface of the midrib. Sterile and female plants are 6–11 mm wide and male plants 2.5–3 mm wide, pseudodichotomously furcately branched.

Cells of margin on dorsal side very variable from almost isodiametric (40–)50 × 50 μm to slightly 30–35(–40) × 50–55 μm or two-three times and than 25–35 × 50–85 μm elongated. Thallus in cross section are 8–12 cells high medially, gradually thinner towards margin into the more or less undulate wings, that are in 1- several rows one cell thick, cells of

medula in cross section thin walled lacking thickening bands, variable from single just 25 × 25 but mostly large (35–)50–75 × 60–90 μm, epidermal cells elongated 20–25(–30) × (30–)35–40(–50) μm. Apex of thallus on ventral side with slime-hairs of 2–8 elongated cells varying from 37 × 60 to 30 × 125, so 1.5–4.5 times as wide as long, often not clearly visible on dry material. Vegetative propagation in autumn, by repeatedly pseudodichotomously branched, light green (even when dry) fragile thallus apices.

Dioicous. Male and female plants often in mats close to each other or intermingled. Perichaetium cylindrical, with lobulate mouth with mostly two or more cells wide relatively short cilia, sometimes with single one-cell wide cilia of several cells long (up to 5). Antheridia located on dorsal side along midrib, each in small cavity surrounded by elevated tissue with open crater-like top in one, two and sometimes three irregular rows, sometimes in groups of two to three tubercles. The cells surrounding the hole ("crater") of male tubercles are non papillose.

Capsule spherical on very long colorless stalk, 4-valved, outer layer with nodular thickenings. Elaters numerous (2–)3(–4) spiral, spores light brown large ellipsoidal, several-celled, 40–50 × 70–80 μm.

Distribution. *Apopellia endiviifolia* is widespread in temperate regions with calcareous and basic rocks both in mountains (up to 2300 m alt. in Caucasus and 2500 m alt. in the Alps [30]) and lowland in the southern and western Europe with a frequency of occurrence decreasing to the north. The species has also been recorded from the Mediterranean, coastal regions of North Africa, mountains of South Siberia (Altai) and Central Asia (Uzbekistan), China, Korea, Japan, the Himalayas (India) and Oman [30,31]. The northern-most known locality is in Scandinavia at 66°15′ N [32]. Records from Karelia [33] are erroneous and referred to *Pellia neesiana* during this study. A record of *Apopellia endiviifolia* (*Pellia endiviifolia*) from Murmansk Province [34] is referred to *A. alpicola* in this study. Most records from Siberia were referred to other species after revision, but a single sequenced specimen from Altai Mountains is located in the *A. endiviifolia* clade. Records from Japan, China and India and the whole of South-Eastern Asia should be verified for possibly belonging to other species.

Selected specimens examined. **RUSSIA**: Moscow Province, Krasnogorskiy District, spring fen with calcareous tuff, M. Ignatov, 26.04.1986 [KPABG108496, with sporophytes]; Mari El Republic, N056°38′57.00″ E047°13′47.00″ floodplain of Bol'shaya Kokshaga River, on concrete blocks in ditch along railway, Konstantinova, K439-2-04, 14.09.2004 [KPABG 108034] with *Marchantia polymorpha* subsp. *ruderalis*, *Lophocolea heterophylla*; Caucasus, Republic of Adygeya: alpine meadow, at base of rock, in *Rhododendron* tickets, alt. 2182 m, Konstantinova & Savchenko [KPABG 113355, per.], Lagonaki Plateau, sinkhole on the slope of Abadzesh Mnt, 1949 m alt., in water, K437-2-07 Konstantinova & Savchenko [KPABG 115936], Lagonaki Plateau, seepage at base of mountain slope, on dying grasses in bed of creek, alt. 1916 m, K437-2-12 Konstantinova & Savchenko [KPABG 115936], valley of Belaya River, sand covered cliff, alt. 527, Konstantinova & Savchenko, K165-11-09 [KPABG 113382], Krasnodar Territory: valley of Msymta River, moist cliff, alt. 250 m Konstantinova & Savchenko, K173-09 [KPABG 113017], valley of Achipse River, on moist cliffs on bank of waterfall, alt, 1210 m, Konstantinova & Savchenko, K165-11-09; near Lasorevskoe Town, Svirskaya shshel, Konstantinova & Savchenko, K343-4-11 [KPABG 114974]; Chechnya Republic, Shatoiskiy District. alt. 770 m. Doroshina, 8-08-2018 [KPABG 122154]. Altai Republic. Shebalinsky District, Elekmonar Creek valley, 3–5 km upstream from the Katun' River, on fallen log across the creek, 700 m a.s.l., 2 Aug. 1991, M.S. Ignatov 26/11 [MHA-9085521]. **BELGIUM**, Belgian Ardennes, Bévercé near Malmedy, north-exposed slope calcareous conglomerate rocks, ca 350 m a.s.l., on stone of stairs, 01.10.1999, Konstantinova, H.During, & H.van Melick [KPABG102128].

Ecology. The species occurs in moist places in deep shaded ravines mainly on shaded moist limestone, siltstone cliffs and rocks and sometimes on the base of trees, on moist soil in the thickets of bushes on banks of small streams, on rocks in the drying river beds, on calcareous conglomerate rocks on steep slopes in the forest along streams and on moist

calcareous rock outcrops, on cliffs on roadsides in ravines, in rich fens, along seepages, on rocks in moist fir-beech forests, on soil along roads trails and on concrete blocks along railways. It is a widespread species, often occurring in huge pure mats or with admixture of calciphiles or basiphiles such as *Jungermannia atrovirens*, *J. pumila*, *Mesoptychia badensis*, *Preissia quadrata*, *Conocephalum conicum*, *Marchantia paleacea*, *Asterella lindenbergiana*, *Southbya tophacea* or neitrophiles like *Chiloscyphus polyanthus*.

*Apopellia megaspora* (R.M. Schust.) Nebel & D.Quandt [*Pellia megaspora* R.M. Schust. In J. Bryol. 11: 419, Figures 1 and 2. 1981—Holotype: USA, Massachusetts, Gorge of Green River, Colrain, Franklin County, ca. 110 m a.s.l. *Schuster 73-109* (F n.v.). Topotype: Konstantinova & Schuster, 27.05.1992 [KPABG 123479]. Figures 6 and 7.

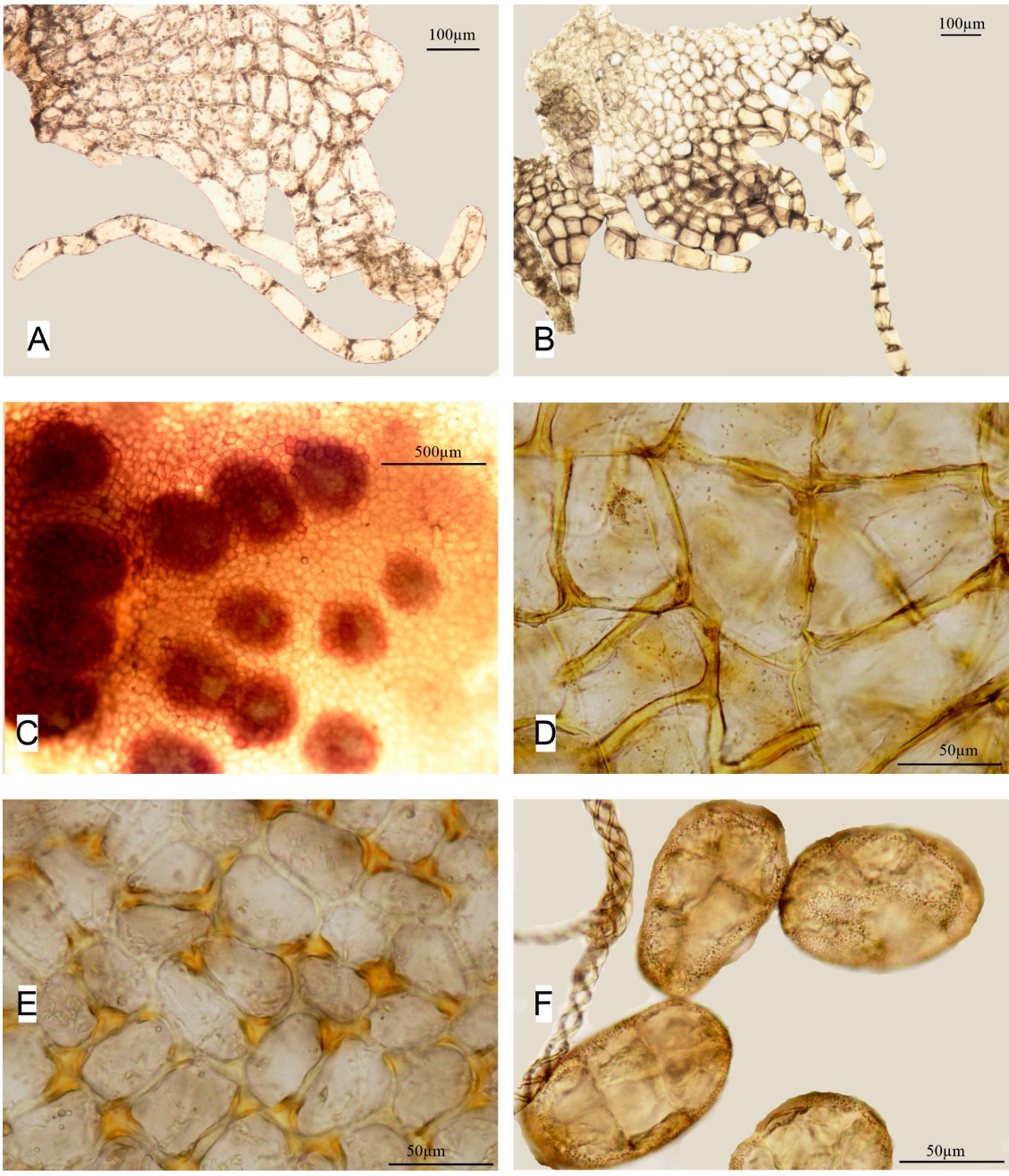

**Figure 6.** Apopellia megaspora: (**A**,**B**)—perianth mouth; (**C**)—group of antheridia cavities; (**D**)—capsule inner wall cells; (**E**)—capsule epidermal cells; and (**F**)—spores & elaters; (**A**,**B**) from KPABG 123479; and (**C**–**F**) from KPABG-116783).

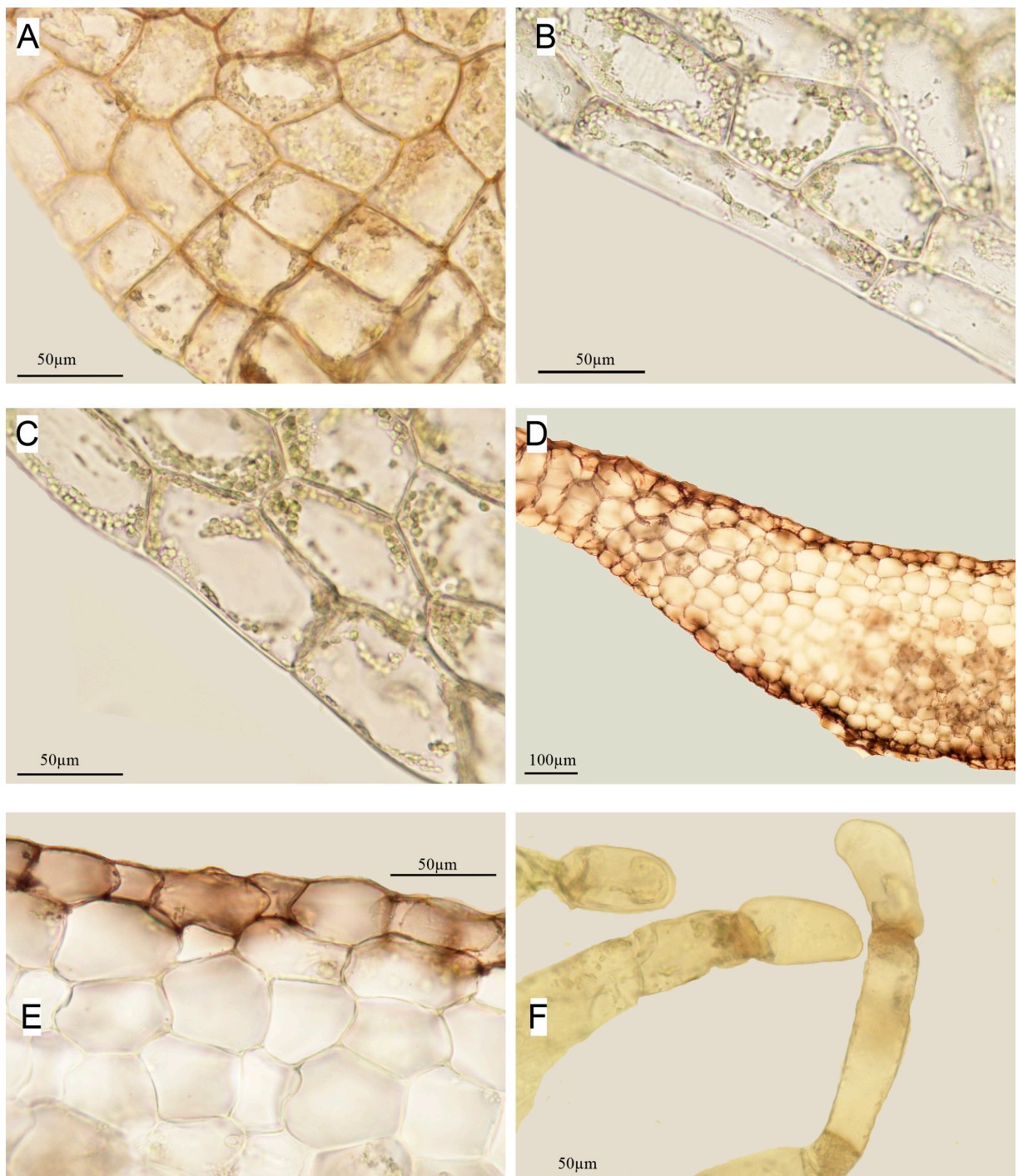

**Figure 7.** *Apopellia* megaspora: (**A**–**C**)—marginal cells; (**D**)—part of cross-section;(**E**)—part of cross-section showing small cells of epidermis contrasting with large cells medula; and (**F**)—slime hairs (all from KPABG 123479).

Thalli dark green, translucent to almost black in herbarium (in specimen from Taimyr Peninsula), 5–8 mm wide and 2-several cm long, thin, marginal cells of thalli very variable from almost isodiametric 30 × 30 to 1.5–2–3(–4) times as long as wide and 25–35 × 50–75 (–100) µm in several rows, one celled and then gradually turning into multi-layered costa 8–12 cells thick, cells in crossection thin-walled mostly large 35–40(–50) × 37–65 µm with single small cells ca. 25–30–35 µm, cells of epidermis (20–)25–30 ×30–50 µm. Costa on ventral side can in some plants be slightly red-brown colored. Slime hairs on ventral side

numerous near the apex and scattered below red-brown colored or colorless and then hardly visible, consisting of 3–8 distinctly elongated cells ca. (25–)30–45 × 50–200[400–800] μm with distinctly thickened upper wall of apical cell. Vegetative propagation not known.

Dioicous. Male plants narrow, in separate mats but close to female or in the same mat, antheridia in unclearly defined groups of numerous tubercles in one or two groups on different branches, when young not differing in color from thallus but later becoming dark red brown. Antheridia large, one per tubercle. Perichaetia short cylindrical, heavily inciste-ciliate with one seriate ending cilia up to 20 elongated cells length, cells of cilia (35–)40–60 × 75–150 μm.

Capsule spherical on very long colorless stalk, 4-valved, 3–4 stratose. Epidermal cells subquadrate to polygonal, ca. (25–)30–40 × 30–50 μm with nodular angles. The innermost layer consists of large irregular cells ca. (35–)45 × 50(–60) × 50–75(–90) μm without distinct semiannular bands but with some thick walls. Spores faintly granulate, light brown with greenish tint of several (to 9) cells, large (63–)68 × 80–100–110[–120] μm. Elaters numerous, 10–14 μm in diameter, (2–)3–4 spiral, elater band 1.5–2 μm wide.

Selected specimens examined. **USA**: Massachusetts Gorge of Green River, Colrain, Franklin Co. Konstantinova & Schuster, 27.05.1992 [KPABG-123479, and., per.], topotype. **RUSSIA**: Arkhangelsk Province, Pinega Nature Reserve, Krasnye Gory Tract, 64.646733 N. 42.902703 E, on clay soil near gypsum cliff wall, M.S. Ignatov 2.VIII.1988 [MHA-9085547]. Republic of Komi, Subpolar Urals, left bank of Bol'shoy Patok River, 64°31′13.4″ N, 58°33′26.1″ E, north faced limestone rock outcrops in valley of brook, flooding zone, 11.06.2013, Dulin s.n. [KPABG-116783, spor.]. Republic of Tuva, Todzhinskaya basin, west bank of Kadysh Lake, 52°35′26.3″ N, 97°04′20.5″ E, moist rock outcrops, 30.08.1999, Otnyukova [KPABG-100790, per.]. Krasnoyarsk Territory, Ereechka River valley, 71°15′ N, 105°37′ E, 190–220 m a.s.l., flood plain, on silty alluvium, 24.07.2013, Fedosov 13-3-0420 [KPABG-116907, and., per.]. Republic of Buryatia, Khamar-Daban Ridge, valley of Pereemnaya River, 7.08.2001, Konstantinova [KPABG-102436, per.]; Barguzin Range, Ulyugna River valley, on soil on the river bank, 55°00′45.4″ N, 110°49′31.1″ E, 931 m a.s.l., 27.06.2016, Mamontov 575-1-5085 [MHA-9085552, per.], Mamontov 575-1-5164 [KPABG-124861, per.]; Ulyugna River valley, on soil on the river bank, 26.06.2016, Mamontov 574-1-5156 [MHA-9090615]; Nyundyu River basin, Duvan Pass, 55°04′25.1″ N, 110°37′12.9″ E, 1713 m a.s.l., on moist soil in gorge, in partial shade, 02.07.2016, Mamontov 601-6-5046 [MHA-9090617]. Trans-Baikal Territory, Stanovoy Highlands, Kodar Range, Syul'ban River valley, 56°50′12.1″ N, 117°17′26.1″ E, 1600 m a.s.l., in deep rock niche, in partial shade, 13.06.2015, Mamontov 533-6-7183 [MHA-9090619].

Ecology. The species was described as "decidedly calciphilous" [7]. Data obtained by us support this statement. All findings in Siberia and Urals are restricted to the areas with distribution of calcareous rocks as well. The species occurs on peat banks of lakes and rivers mostly in flooding zone or in rock outcrops on banks of lakes. It grows in pure mats or mixed with *Moerckia flotoviana, Aneura pinguis, Mesoptychia* spp. e.g., in Ereechka River valley (Taimyr Peninsula) where it occurs in area with "white Proterozoic siliceous limestones & dolomites" [35] as *Pellia endiviifolia*. The species occurs in mountains of South Siberia both on soil on banks of streams, as well as on moist soil and even moist logs in flood land forests or ravines, sometimes in communities with relictual vascular plants, e.g., *Rhododendron aureum* with such calciphiles as *Mesoptychia badensis, M. gillmanii, Preissia quadrata*.

Distribution. Schuster characterized this species as "strictly northern plant, absent or at least exceptionally rare, south of Massachusetts and New York" [7]. The species is assuredly not rare in eastern North America, with northernmost localities in Labrador and Newfoundland (l.c.). It has recently been recorded in western North America [11] and in this study. During this study, the species was found in several localities in the north-eastern part of Europe (European part of North Urals, Arkhangelsk Province) and many stations in Siberia, both in the mountains of South Siberia and mountains of the north of East Siberia (Figure 3). In the mountains of southern Siberia, the species reaches a height of

1713 m alt. and in the north of East Siberia the species occurs far beyond the Arctic Circle at ca. 71°15′ in Taimyr Peninsula. The species is most likely more widespread, but to find out its distribution requires a critical revision of all collections, especially American and Asian, as well as targeted collection of specimens with sporophytes, which will greatly facilitate identification.

*Key to the Species of Apopellia*

1. Plants with massive costa (up to one third of the width of the thallus) mostly red- to purple-colored on ventral side with perichaetia more or less horizontal with distinctly uneven sides: short and strongly lobulate-long ciliate anteriorly and much longer and slightly lobulate posteriorly, antheridia in groups, one per plant or one on each branch. Plants restricted to acidophilous or neutral sites in midlands to high mountains or high latitudes ..........................................................,................................................*Apopellia alpicola*

1. Plants with tubular perichaetium lobulate-ciliate at apex. Plants mostly restricted toCa-richsubstrates..........................................................................................................................2

2. Plants with perichaetia erect or subhorizontal tubular or cylindrical with many acute and mostly short ciliate-dentate lobes, antheridia scattered along the costa in two-three irregular rows, spores 35–45 × 70–80 μm, plants in autumn with characteristic fragile repeatedly pseu-dodichotomously branched, thallus apices............................................... *Apopellia endiviifolia*

3. Plants with antheridia in large groups, perichaetia with acute and long ciliate-dentate lobes, spores very large 70–80 × 90–110 μm, plants without autumn prolifera-tion..........................................................................................................*Apopellia megaspora*

## 6. Conclusions

Our research supports the treatment of *Apopellia* as a separate genus and *A. alpicola* as a distinct species. We radically changed the idea of the distribution of species of the genus. It was found that *A. megaspora* and *A. alpicola* are much more widespread than previously assumed, being almost circumpolar species. In addition, it became obvious that we are dealing with sympatric distribution of *A. megaspora* and *A. alpicola*. Both species were found in adjacent areas at least in the Alberta Province of Canada, and in the Republic of Tuva and Trans-Baikal Territory of Russia. *A. megaspora* is a strictly calciphilous species, whereas *A. alpicola* always occurs in neutral or even acid conditions; *A. endiviifolia* is more or less indifferent, occurring either on neutral, basic or Ca-rich substrates, but in general in more dry and warm climates. In Eurasia, all three species are reliably known, while we have no evidence yet of occurrence of *A. endiviifolia* in America. Whether this is true or not, further research will show. It is obvious that it is necessary to purposefully collect *Apopellia* species with sporophytes or perianths, which will allow them to be determined more or less reliably. It is probably trustworthy enough to assign species with autumn proliferations to *A. endiviifolia.* At the same time, we did not find any data supporting occurrence of *A. endiviifolia* in America except for a note in a letter from D.H. Wagner to one of the authors (Anna Vilnet), saying that "Thallus proliferations occur occasionally in Oregon material". All three species probably occur in western North America as in Siberia. We are obviously still far from finding out the true distribution of them. It is necessary to revise material (especially from mountain systems in central and eastern Asia and America) stored in herbariums based on the descriptions provided by us here. An especially important task for the future is the collection and study of plants with sporophytes, or at least perichaetia, to avoid misidentification that allows us to clarify the distribution, ecology and taxonomy of species.

The modern ranges of species, together with the postulated antiquity of the genus [5], suggest their relict character. However, due to the apparent lack of knowledge of the distribution of species, we have so far refrained from specific conclusions.

**Author Contributions:** Conceptualization, N.A.K.; methodology, N.A.K. and A.A.V.; formal analysis, N.A.K., A.A.V. and Y.S.M.; investigation, N.A.K., A.A.V. and Y.S.M.; resources, N.A.K. and Y.S.M.; data curation, N.A.K., A.A.V. and Y.S.M.; writing—original draft preparation, N.A.K. and A.A.V.; writing—review and editing, N.A.K., A.A.V. and Y.S.M.; visualization, N.A.K. and A.A.V.; supervision, N.A.K.; and project administration, N.A.K. All authors have read and agreed to the published version of the manuscript.

**Funding:** The study was carried out within an institutional research project of the Avrorin Polar-Alpine Botanical Garden-Institute, RAS NN No. 1021071612832-8-1.6.11, and using large-scale research facilities of the "Herbarium of the Polar-Alpine Botanical Garden-Institute (KPABG)", reg. No. 499397. The study by Yu. S. Mamontov was supported by the Russian Ministry of Higher Education and Science for grant 075-15-2021-678 supporting Center of Common Use «Herbarium MBG RAS» and within the State assignment (122042700002-6).

**Institutional Review Board Statement:** Not applicable.

**Data Availability Statement:** The data supporting reported results can be found in herbarium KPABG and in the "L" Information system https://isling.org/ (accessed on 23 July 2023).

**Acknowledgments:** We sincerely thank A. Savchenko for preparing the distribution maps and assistance in processing photos for publication. A. Hagborg is gratefully acknowledged for many useful comments and linguistic corrections.

**Conflicts of Interest:** The authors declare no conflict of interest.

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
