# Peer review of "The Current Diversity and Distribution of the Simple Thalloid Genus Apopellia (Marchantiophyta): Evidence from an Integrative Taxonomic Study"

_diversity, doi:10.3390/d15080887_

Round 1

Reviewer 1 Report

+//

This paper presents the results of a detailed and meticulous morphological and molecular study of the genus Apopellia. The new data provide a sound basis for identifying the different taxa and their global distributions. This work will be widely read and welcomed by bryologists. All the methodologies appear sound and the conclusions well argued.

Minor revisions to the English are recommended.

Change the title  primitive to simple is better

The text would benefit from more commas and some of the English is rather awkward. eg last sentence of Abstract

Androecia of Apopellia alpicola were by microphotographs of the more important morphological features.

Change to     Microphotographs illustrate the more important---

Intro para 1.    All American specimens involved in the molecular study by Schütz et al. [1] were attributed

Change to;-  All American specimens used in the molecular study by Schütz et al. [1] were attributed

Intro para 2. Morphological differences, at least for A. megaspora, are quite clearly expressed in the generative sphere, which is reflected in the name of the species.

Change to;- Morphological differences, at least for A. megaspora, are clearly expressed in the spores hence the name of this species.

Intro last para     Schuster described this species and, thus, get a DNA loci of A. megaspora from the site shown by Schuster as the type locality. Another goal was to figure how to treat the Western North American specimens, which, based on their distribution, ecology and morphology, can be attributed to A. alpicola. Once more task was to find out whether A. megaspora really occurs in the Asian part of Russia, as maintained by Schütz et al. [1], in which the sequenced specimen from Turukhansk, valley of the Yenisei River Krasnoyarsk Territory (not Lena, where the authors mistakenly placed it) turns out to be in one of the subclades of A. megaspora along with Western North American specimens. Finally, we want to clarify the distribution of the genus based on available data.

Change to;- e Schuster described this species and, thus, obtain DNA loci of A. megaspora from the site shown by Schuster as the type locality. Another goal was to consider how to treat the Western North American specimens, which, based on their distribution, ecology and morphology, can be attributed to A. alpicola. A further task was to find out whether A. megaspora really occurs in the Asian part of Russia, as maintained by Schütz et al. [1], in which the sequenced specimen from Turukhansk, valley of the Yenisei River Krasnoyarsk Territory (not Lena, where the authors mistakenly placed it) turns out to be in one of the subclades of A. megaspora along with Western North American specimens. Finally, we clarify the global distribution of the genus.

2,2. Plants with perianth and sporophytes were studied especially carefully.

Change to;-  . Plants with perianths and sporophytes were studied in detail.

Another goal was to consider how to treat the Western North American specimens,P2  Another goal was to figure how to treat the Western North American specimens, which, based on their distribution

Change to;-   Another goal was to consider how to treat the Western North American specimens,

Finally, we want to clarify the distribution of the genus based on available data.

Change to;-  Finally, we clarify the global distribution of the genus

P14   The data obtained shows that   Data show   plural[JD1] 

 [JD1]

This just needs minor editing

Author Response

We thank the reviewer for the helpful comments and very useful suggestions about english and corrected the manuscript in accordance with them.

  1. We agree that simple is better than primitive and corrected it.
  2. All recomended changes in english were accepted.

Reviewer 2 Report

The manuscript deals with taxonomic problems of the liverwort genus Apopellia (Marchantiophyta). The authors found additional support from nuclear DNA for the treatment Apopellia as a separate genus. The manuscript is quite well-structured and illustrated. The conclusions are based on the Results. However, some corrections are needed.

Comments and suggestions the manuscript

Introduction

The Introduction would be more complete and deeper if the first paragraph of the Discussion were moved to it.

At the end of the Introduction, several objectives of the study are listed, but there is a lack of a clearly defined main research aim.

Materials and methods

When he first time an acronym of Herbarium is mentioned, it is a reasonable  to give the full name of it, or simply indicate that the acronyms follow the Index Herbariorum....

What do the dots on the map represent? Just the coordinates of the samples or a grid cell of a certain size?

Results

What is presented in the Results section 3.2 Morphology, is more appropriate for the Discussion section where there is very little discussion of morphological characters. The Results section would benefit from more emphasis on morphological differences and similarities (possibly in the form of a table). So, this part should be signifficantly extended.

Section 3.3 Distribution should also highlight the points that have been obtained as a result of the research and that are appropriate for discussion. This part should be also signifficantly extended. It deals exceptionally distribution of the species, nothing about ecology.  

 Discussion

My suggestion– first paragraph of the Discussion would be good for Introduction.

Author Response

Dear reviewer,

Many thanks for your suggestions. We corrected manuscript in accordance with your comments.

Introduction

We mooved the first paragraph of the Discussion to Introduction and add at the beginning of the last paragraph the main goal of the paper.

2. Material and methods

We have added the full name of herbariums, leaving the acronyms in parentheses.

We have explained what the points on the map are based on by adding the following sentence:

The points on the maps are placed according to the coordinates available in the labels or calculated from the description of the locations.

Results 

We mooved section 3.2 Morphology to Discussion as it was suggested by reviewer, but we believe that it is no need to expand the results by "morphological differences and similarities". It is given in the section Taxonomical treatment as well as in provided key to species.  

Section 3.3 

We have removed the word ecology from the subtitle, as we have not conducted any special studies on ecology, and the label data give very incomplete information, which we have given in the descriptions of the ecology of each species.

Reviewer 3 Report

Please include the primers You used.

Key to the species must be thesis and antithesis. Please, correct.

Figures and Tables should be carefully checked up. The labels are wrong like in Fig 6. where G does not exist. Similarly in Fig7. F is missing.

Table 2 is not visible!

The English is sometimes hard to follow. It seems it is written by different persons so the quality vary within the Manuscript. Some improvement is needed. In some sentences verbs are omitted.  

Author Response

Dear reviewer, 

Many thanks for your corrections and suggestions.

As to the primers, we provide citation for used primers such we didn't modified them for current study.

We have corrected the key.

We have corrected the captions to the figures. 

In our browser, the table is readable.

The English in some phrases has been corrected in accordance with the proposed version of the English-speaking reviewer

Round 2

Reviewer 2 Report

All the comments made earlier have been taken into account.